# Multimedia Independent Multipath Routing Algorithms for Internet of Things Based on a Node Hidden Communication Model

**Cong Wu [1,\*] and Jianhui Yang [2]**

[1] School of Computer Science and Technology, Zhoukou Normal University, Zhoukou 466001, China
[2] School of Mathematics and Statistics, Zhoukou Normal University, Zhoukou 466001, China; yjh_2019@126.com
\* Correspondence: wucong@zknu.edu.cn; Tel: +86-1352-571-9858

**Abstract:** In order to achieve a multi-path routing algorithm with time delay and energy consumption balance to alleviate the energy holes around a sink, a multimedia independent multipath routing algorithm for internet of things (IoT) based on node hidden communication model is proposed in this paper. On the premise of satisfying the application delay, a multi-source multi-path routing algorithm is proposed by using the idea of software definition and fitting multiple curves to form independent multi-path routing. Through a sink node centralized programming control source node routing, according to the priority of the source node, the dynamic angle of the source node can be allocated, which effectively reduces the energy consumption of the network. In addition, considering that the Internet of Things has more perceptive nodes, limited computing and storage capacity, frequent joining and exiting operations and other factors, a hidden communication model of nodes is designed for the IoT. It is helpful to improve the level of privacy protection in the IoT, and to effectively improve the ability of nodes to resist attacks in the IoT. The experimental results show that the proposed algorithm avoids the interference between paths and various network attacks to the greatest extent, and the energy consumption is relatively low under the requirement of quality of service (QoS) delay.

**Keywords:** independent multipath routing algorithm; the Internet of things; wireless multimedia sensor network (WMSNs); multimedia data processing; node concealment communication model; multimedia security; the network energy consumption

---

## 1. Introduction

Internet of Things (IoT) applications will bring huge amounts of information data processing, including not only text, but also multimedia data such as audio and video. As the multimedia data perception layer of the IoT, wireless multimedia sensor network (WMSNs) has been widely used in various fields, including target monitoring, tracking and environmental monitoring. Multimedia data requires high bandwidth and delay, so it is necessary to establish multipath routing for transmission. Independent multipath routing can improve network throughput and work efficiency and minimizing overlap between multipaths can help WSN nodes to use energy equally to reduce black hole phenomenon.

Resource-constrained wireless sensor nodes are prone to congestion in high-load applications, which significantly affects network performance [1–3]. In order to solve this problem, researchers have adopted more network resources to improve network capacity and adopted the method of multi-path average network traffic to support the bandwidth requirements of different applications to solve the problem of network congestion. Moreover, sending network traffic through multiple sensors will

reduce the average energy consumption of sensor nodes and prolong the network lifetime. In the past decade, multipath routing has been widely used for various purposes, such as improving the reliability of data transmission, providing fault-tolerant routing, congestion control and quality of service (QoS) support. In single-channel wireless networks, because of the broadcast characteristics of wireless communication, sensor nodes use shared wireless channels to communicate with adjacent paths. Intensive concurrent communication may cause interference and increase the probability of packet collision between active path nodes. This problem is called path coupling effect, which limits many of them. This problem poses a challenge to the design of efficient multipath routing protocols. In order to reduce the interference between multipath routing, designing independent and disjoint paths becomes the main standard for designing multipath routing protocols. The main contributions of this paper are summarized as follows:

(1)  On the premise of satisfying the application delay, using the idea of software definition, the independent multi-path routing is formed by fitting multiple curves. A multi-source multi-path routing algorithm is proposed, which takes into account the balance of delay and energy consumption.

(2)  Aiming at the problem of privacy disclosure of communication relationship, a node hidden communication model is designed for the IoT. In this covert communication system, the attack detection and prevention technology is studied, which effectively improves the anti-attack ability.

The structure of this paper is as follows: chapter 2 introduces the related research on Multimedia Routing Algorithm in the IoT. chapter 3 introduces the system model in detail. chapter 4 introduces the design of multimedia independent multipath routing algorithm. chapter 5 introduces the implementation process of multimedia independent multipath routing.

## 2. Relevant Research

With the development of hardware technology, WMSNs have been widely used in various fields, including target monitoring, tracking and environmental monitoring. Wireless multimedia sensor networks introduce multimedia sensors such as audio and video into wireless sensor networks. WMSNs are generally used in real-time applications with high bandwidth and low latency. In recent years, researchers have done a lot of research on routing algorithms, and many routing protocols are proposed according to the performance requirements of different applications. If multimedia data is transmitted through a single path, the energy of the forwarding node will be exhausted rapidly and the node will fail. Multipath routing method constructs multiple paths between source and target nodes, divides data into multiple paths equally, and makes more nodes participate to prolong the lifetime of nodes and networks. Data sent by multipath routing is more secure and reliable than single-path routing [4]. Multipath routing has been widely used in different requirements, such as improving the reliability of data transmission, providing fault-tolerant routing, congestion control and QoS support. Early multipath routing was proposed based on traditional wireless sensor networks (such as ad hoc networks). To be used in wireless sensor networks, it is necessary to consider the unique characteristics of wireless sensor networks (such as limited energy supply, computing power and memory capacity) and short-range wireless communications (such as signal fading and interference) [5–7].

Due to the large amount of multimedia information collected by WMSNs, the energy consumption of data acquisition, processing, storage, and wireless transceiver is relatively large, especially around sink nodes, energy holes are easily formed, which seriously affects the performance of WSN [8]. The network lifetime can be improved by balancing the energy load distribution around sink nodes, and how to alleviate energy holes around sink nodes is also an important research direction.

Nodes can use location technology to obtain their own geographic location information. According to coordinate information, nodes can avoid blind flooding of routing detection packets and efficiently discover and maintain routing, which has good scalability and robustness. This paper mainly studies the multipath routing protocol based on geographic location information. Tarique et al. [9] proposed a

DGR multipath routing protocol solves the problem of bandwidth and energy limitation in real-time video streaming. From a small number of scattered video sensor nodes in wireless sensor networks (WSN), data is transmitted to sink nodes by combining forward error correction (FEC) coding. Shu L et al. [10] offers an algorithm is a two-stage geographic path greedy protocol, which supports the shortest path routing and multipath routing around wormholes. The two phases of TPGF are based on geographic information forwarding and path optimization. Repeated execution of TGPF can find multiple paths to send data. However, it only considers the formation of multipath paths, without considering the interference between paths and traffic distribution. Chen A et al. [11] based on the improvement of SPEED protocol, SPEED protocol maintains a transmission speed throughout the network, which is not suitable for various data types of wireless multimedia sensor networks. RTGOR designed different QoS services in terms of real-time and reliability. Aswale et al. [12] proposed a geographic multipath routing (TIGMR) protocol based on triangular link quality metrics and minimum path interference, which discovers disjoint paths of multiple nodes in the IEEE 802.15.4 compatible network. The cross-layer routing protocol selects forwarding nodes based on triangular link quality metrics, residual energy and distance, and predicts the smallest adjacent path interference effect.

The above independent multipath routing algorithms are based on the maximum transmission distance per hop, without considering the balance of delay and energy consumption. In WMSNs applications with lax delay requirements, they can reduce the distance per hop and save energy consumption, in this section, delay and energy balance independent multipath routing algorithm is studied; When multiple source nodes send data to sink nodes through multiple paths at the same time, in order to avoid interference caused by path crossover, multiple source nodes need to coordinate transmission paths with each other. At present, there is little research on multi-source and multi-path routing. In this paper, a new multi-source and multi-path routing algorithm is proposed by applying the idea of software definition and controlling the routes of multiple source nodes centrally through sink nodes.

## 3. System Model

### 3.1. Delay Model

Each hop delay consists of four parts, ①queuing delay, the queuing delay can be neglected if unsaturated traffic is used; ②processing delay, Assuming that each node processes a constant length packet with equal delay; ③propagation delay, it can be neglected compared with other parameters; ④transmission delay, generally, with constant unsaturated traffic, the size of the packet remains unchanged, and the node does not need to wait. Therefore, the delay per hop can be simply estimated as a constant, and the end-to-end delay is proportional to the number of hops. Based on the data in reference [13], the delay of 802.11b DCF per hop is estimated. The delay $T_{hop}$ of successful data transmission includes transmission delay $T_{DATA}$ and response delay $T_{ACK}$:

$$T_{DATA} = T_{PRE} + T_{PHY} + \frac{8L_{MAC} + 8L_{DATA}}{R_{DATA}} \tag{1}$$

$$T_{RTS} = T_{PRE} + T_{PHY} + \frac{8L_{RTS}}{R_{CTRL}} \tag{2}$$

RTS/CTS delay can be calculated by the following formula:

$$T_{RTS} = T_{PRE} + T_{PHY} + \frac{8L_{RTS}}{R_{CTRL}} \tag{3}$$

$$T_{CTS} = T_{PRE} + T_{PHY} + \frac{8L_{CTS}}{R_{CTRL}} \tag{4}$$

$$T_{hop} = T_{RTS} + \tau + T_{SIFS} + T_{CTS} + \tau + T_{SIFS} + T_{DATA} + \tau + T_{SIFS} + T_{ACK} + \tau + T_{DIFS} + \frac{CW_{min}\sigma}{3H} \quad (5)$$

The symbols and values used are shown in Table 1. According to Table 1, the value of $T_{hop}$ with RTS/CTS is calculated to be 2.68 Ms. If the end-to-end delay $T_{QOS}$ is specified, the delay $H_{QOS}$ per hop can be calculated by the following formula:

$$H_{QOS} = \left\lfloor \frac{T_{QOS}}{T_{hop}} \right\rfloor \quad (6)$$

**Table 1.** Calculating delay symbols and values per hop.

| Symbol | Definition | Value |
|--------|-----------|-------|
| T | Propagation delay | 1 µs |
| $\sigma$ | Time slot | 20 µs |
| $CW_{min}$ | Minimum Competition Window Size | 30 |
| $T_{PRE}$ | Physical Layer Synchronization Code Transmission Time | 128 µs |
| $T_{DIFS}$ | DCF frame interval | 50 µs |
| $T_{SIFS}$ | Minimum Interframe Interval | 10 µs |
| $T_{PHY}$ | Physical Layer Transport Time | 45 µs |
| $L_{DATA}$ | Packet Length | 125 bytes |
| $L_{MAC}$ | MAC Packet Length | 28 bytes |
| $L_{ACK}$ | ACK Packet Length | 14 bytes |
| $L_{BTS}$ | RTS Packet Length | 20 bytes |
| $L_{CTS}$ | CTS Packet Length | 14 bytes |
| $R_{DATA}$ | Data transmission rate | 1 Mbps |
| $R_{CTRL}$ | Control message sending rate | 1 Mbps |

### 3.2. Energy Consumption Model

Energy consumption includes four parts: ①Idle energy consumption: Nodes in the idle state consume little energy and can be ignored. ②Transmission energy consumption: Sending energy consumption accounts for the main part of energy consumption, which is related to the number of bytes sent. ③Receiving energy consumption: It is related to the number of bytes received. ④Control signal energy consumption: Assume that each node is the same.

If the size of the packet is unchanged and the transmission distance is the same, the energy consumption will be the same. Therefore, if a packet is sent from source to sink, and each hop is the largest distance transmission, and the free space transmission path index is 2, that is, the energy consumption per hop is proportional to the square of distance, then the energy model of each hop transmission is:

$$E_{hop} = C.D^\alpha \rightarrow E_{hop} = C.D^2 \quad (7)$$

where $C$ is a constant and $D$ is a hop distance. $\alpha$ is the path loss index, which depends on the transmission environment, free space transmission is generally 2. For simplification, set $C$ to 1 and $\alpha$ to 2. Packets send energy $E$ through $k$ paths $P_k$ from source to target:

$$E = \sum_{i=1}^{P_k} (D_{n_{i-1}n_i})^2 \quad (8)$$

where $D_{n_{i-1}n_i}$ is the transmission distance from $n_{i-1}$ to $n_i$ hops. Then $n_0$ is the source node and $n_{P_k}$ is the sink node.

### 3.3. Transmission Model

After specifying an application delay of $T_{QoS}$, when $k$ paths are generated between Source and Ink, and the end-to-end delay control is close to $T_{QoS}$, the energy consumption of each path is the smallest. $D_{st}$ represents the shortest path, then $T_{QoS}$ needs to satisfy $T_{QoS} \geq \frac{D_{st}}{R_{max}} \times T_{hop}$, otherwise there will be no path to satisfy q's needs. On the other hand, Q can't be too big. For a given network topology, multimedia applications, set the w range:

$$\frac{C.D_{st}}{R_{max}} \times T_{hop} \geq T_{QoS} \geq \frac{D_{st}}{R_{max}} \times T_{hop} \tag{9}$$

where $C$ is constant, generally set to 2, $R_{max}$ is the maximum transmission distance. Assuming that $R^*$ is the average minimum distance per hop, there are:

$$R^* \geq \frac{D_{st}}{H_{QoS}} \tag{10}$$

When constructing $k$ paths, the optimal transmission distance $R_k$ satisfies the following formula:

$$R^* \leq R_k \leq R_{ma} \tag{11}$$

Suppose that for multiple paths of $P_1, P_2, \cdots, P$, the corresponding lengths are $L_1, L_2, \cdots, L_k$. The average transmission distance per hop of the $k$-th path is:

$$R_k = \frac{L_k}{H_{QoS}} \tag{12}$$

$R^* \times H_{Qos} \leq L_k \leq R_{max} \leq H_{Qos}$ can be obtained, where $L_{max} = R_{max} \times H_{Qos}$.

The energy consumed by the $k$-th path is $E_k = \sum_{i=1}^{H_{QoS}} (Dn_{i-1}n_i)^2$, which can be obtained by applying Cauchy inequality.

$$E_k = \sum_{i=1}^{H_{QoS}} (Dn_{i-1}n_i)^2 \geq \frac{\left( \sum_{i=1}^{H_{QoS}} (Dn_{i-1}n_i) \right)^2}{H_{QoS}} \approx \frac{L_k^2}{H_{QoS}} \tag{13}$$

The condition of the above formula is $D_{n_0 n_1} = D_{n_1 n_2} = \ldots = D_{n_{H_{QoS}-1} n_{H_{QoS}}} \approx \frac{L_k}{H_{QoS}} R_k$. That is to say, when the distance per hop is equal ($R_k$), the energy consumption is the smallest.

### 3.4. Node Hidden Communication Model

3.4.1. Correctness Analysis of Communication Model

This paper introduces a model of node hidden communication, which consists of three kinds of nodes: source node composed of sender $S$, destination node composed of receiver $D$ and relay node (Relay node $R_{(i,j)}$ ($i = 1, 2, \cdots, m; j = 1, 2, \cdots, n$)). The entry node of the forwarding network is the first row of forwarding relay nodes and the next hop of the source node, which is represented by $R_{(i,j)}$ ($i = 1, 2, \cdots, m; j = 1, 2, \cdots, n$); the outlet node of forwarding network is the last relay node in its path and the last hop of destination node, which is represented by $R_{(i,j)}$ ($i = 1, 2, \cdots, m; j = n$).

In the data forwarding network based on network coding, assuming that there is no transmission error probability, all network coding information, local coding matrix and global coding vector existing in $m$ paths can be transmitted to the destination node, through these information, the target node can correctly recover the information sent by the source node [14,15].

After $n$ network coding, information fragmentation $M_i$ can be transformed into information $M_i^{(n)}$, the following is the specific transformation process:

$$
\begin{aligned}
M_i^{(n)} &= C_{(i,n)} \oplus M_i^{(j-1)} = C_{(i,n-1)} \oplus M_i^{(j-2)} \\
&= C_{(i,n)} \oplus \left( C_{(i,n-1)} \oplus \left( C_{(i,n-2)} \oplus M_i^{(j-3)} \right) \right) \\
&= C_{(i,n)} \oplus \left( C_{(i,n-1)} \oplus \left( C_{(i,n-2)} \oplus \left( \cdots \left( C_{(i,0)} \oplus M_i \right) \right) \right) \right) \\
&= \bigoplus_{k=0}^{n} C_{(i,k)} \oplus M_i
\end{aligned}
\tag{14}
$$

The global coded vector ciphertext $EV_i^{(0)}$ is calculated by $n$ times, and then the ciphertext $EV_i^{(n)}$ of the global encoding vector can be obtained:

$$
\begin{aligned}
EV_i^{(n)} &= \sum_{k=1}^{n} C_{(k,n-1)}^{k} EV_i^{(n-1)} \\
&= \sum_{k=1}^{n} C_{(k,n-1)}^{k} \left( \sum_{k=1}^{n} C_{(k,n-2)}^{k} EV_i^{(n-2)} \right) \\
&= \sum_{k=1}^{n} C_{(k,n-1)}^{k} \left( \sum_{k=1}^{n} C_{(k,n-2)}^{k} \left( \sum_{k=1}^{n} \cdots \left( \sum_{k=1}^{n} C_{(k,1)}^{k} EV_i^{(0)} \right) \right) \right) \\
&= \sum_{k=1}^{n} C_{(k,n-1)}^{k} \left( \sum_{k=1}^{n} C_{(k,n-2)}^{k} \left( \sum_{k=1}^{n} \cdots \left( \sum_{k=1}^{n} C_{(k,1)}^{k} EC_{(1,0)}^{i}, EC_{(2,0)}^{i}, \cdots, EC_{(m,0)}^{i} \right) \right) \right) \\
&= \left( \bigoplus_{k=0}^{n-1} EC_{(1,1)}^{i}, \bigoplus_{k=0}^{n-1} EC_{(1,2)}^{i}, \cdots, \bigoplus_{k=0}^{n-1} EC_{(1,k)}^{i} \right)
\end{aligned}
\tag{15}
$$

By decrypting the secret key, the target node can decrypt the global encoding vector's ciphertext $EV_i^{(n)}$, and get the global encoding vector $V_i^{(n)}$:

$$
\begin{aligned}
EV_i^{(n)} &= \sum_{k=1}^{n} C_{(k,n-1)}^{k} EV_i^{(n-1)} \\
&= \sum_{k=1}^{n} C_{(k,n-1)}^{k} \left( \sum_{k=1}^{n} C_{(k,n-2)}^{k} EV_i^{(n-2)} \right) \\
&= \sum_{k=1}^{n} C_{(k,n-1)}^{k} \left( \sum_{k=1}^{n} C_{(k,n-2)}^{k} \left( \sum_{k=1}^{n} \cdots \left( \sum_{k=1}^{n} C_{(k,1)}^{k} EV_i^{(0)} \right) \right) \right) \\
&= \sum_{k=1}^{n} C_{(k,n-1)}^{k} \left( \sum_{k=1}^{n} C_{(k,n-2)}^{k} \left( \sum_{k=1}^{n} \cdots \left( \sum_{k=1}^{n} C_{(k,1)}^{k} EC_{(1,0)}^{i}, EC_{(2,0)}^{i}, \cdots, EC_{(m,0)}^{i} \right) \right) \right) \\
&= \left( \bigoplus_{k=0}^{n-1} EC_{(1,1)}^{i}, \bigoplus_{k=0}^{n-1} EC_{(1,2)}^{i}, \cdots, \bigoplus_{k=0}^{n-1} EC_{(1,k)}^{i} \right)
\end{aligned}
\tag{16}
$$

Through the above three information, the original information can be restored fragmentally, the specific calculation method is as follows:

$$
\begin{aligned}
M_i^{(n)} &= \bigoplus_{k=0}^{n} C_{(i,k)}^{i} \oplus M_i \\
&= \bigoplus_{k=0}^{n-1} C_{(i,k)}^{i} \oplus C_{(i,n)} \oplus M_i \\
&= V_i^{(n)} \oplus C_{(i,n)} \oplus M_i
\end{aligned}
\tag{17}
$$

According to the above formulas, the following calculations can be used to derive the following conclusions:

$$
M_i = M_i^{(n)} \oplus \left( V_i^{(n)} \oplus C_{(i,n)} \right)
\tag{18}
$$

According to $M_i$, the original information $M = (M_1, M_2, \cdots, M_n)$ can be restored.

To successfully construct an anonymous forwarding network between the sender and the receiver of information, it is necessary to satisfy the anonymity of the communication relationship between the sender and the receiver of information.

In the forwarding network, the future self-organizing anonymity of cities and counties can generate the anonymity of the senders of cities and counties with multiple virtual source nodes through the use of pseudonym strategy; the anonymity of the receiver can be realized by broadcasting information from the exit node to the target node; the anonymity of communication relationship between them can be realized through anonymous relay and forwarding network, In each hop of the forwarding network, a group of nodes is included. Each group of nodes can only understand the specific information of the group nodes of the previous and next hop, and can communicate between the group nodes, thus realizing the anonymity of the communication relationship [16,17]. Figure 1 shows the process of constructing forwarding network and the related anonymity strategy.

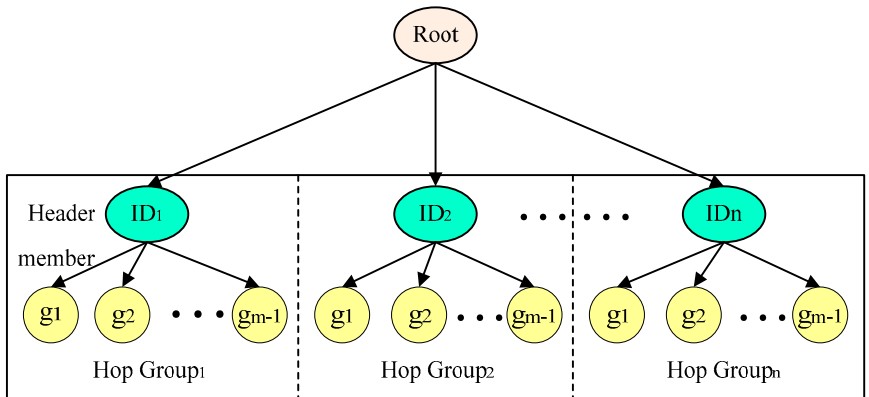

**Figure 1.** Group-based self-organizing anonymous forwarding network.

### 3.4.2. Anonymity Analysis

In the anonymous forwarding network, the sender sends a pseudonym *S*, and only one precursor node exists in the collection of the entry nodes. Although it is easy to infer the sender, the inference is only a pseudonym, and the sender's true identity is not leaked, and the sender's anonymity is guaranteed.

In an anonymous forwarding network, the recipient's exit node is also its precursor node, and there are *m* in subsequent nodes, so the probability that the recipient is inferred by the exit node is only 1 m. The anonymity of the receiver is guaranteed.

In the process of constructing anonymous forwarding network, for each relay node, its information range only includes the last hop and the next hop node, and the information of other nodes is not known. In this situation, even if the attacker can intercept routing information, it is also impossible to infer the relevant information of the receiver and sender. Anonymity of communication relationship is guaranteed.

In the node hidden communication model, the anonymity of the forwarding network, sender, and receiver determines the anonymity of the model. The use of recipient pseudonym ensures that the identity of the node cannot be guessed, and the sender has the anonymity of $d_S = 1$; The number of *n* hop adjacent points determines the anonymity of the receiver, so the sender has the degree of $d_R = \frac{m-1}{m}$ anonymity; The number of forwarding nodes and the length of forwarding path determine the anonymity of forwarding network. If an attacker wants to infer the communication relationship between the receiving and sending sides of the information, he must have a detailed grasp of all the *n* forwarding paths. Combining the probability of an attacker, one of the forwarding paths can be obtained, and the expectation of *k* routing and forwarding paths can be expressed by $E = kp = \frac{k}{n^m}$. The forwarding network has $d_N = 1 - \frac{k}{n^m}$ anonymity. Thus, it can be concluded that the hidden communication path of the whole node has $d = d_N \times d_R \times d_N = 1 \times \left(1 - \frac{k}{n^m}\right) \times \frac{m-1}{m} = \frac{(n^m - k)(m-1)}{mn^m}$ anonymity.

## 4. Design of Multimedia Independent Multipath Routing Algorithms

The algorithm is considered in three steps: ①Calculate the number of hops. Based on each hop delay and application specified delay, the required initial hops are calculated. ②Generate multipath routing, calculate the length of each path, divide the length by hops, and get the distance of each hop. ③In the process of data transmission, the distance per hop can be continuously modified according to the residual delay and the length of the remaining path.

Since the length of each multipath is different, the optimal transmission distance of each hop calculated is also different. That is to say, the node with longer path length has longer transmission distance per hop, while the node with shorter path has shorter transmission distance per hop [18]. Five paths are shown in Figure 2. The longest two paths have a large transmission distance per hop, while the middle one has the smallest transmission distance per hop. However, they have the same number of hops, so the transmission delay is the same.

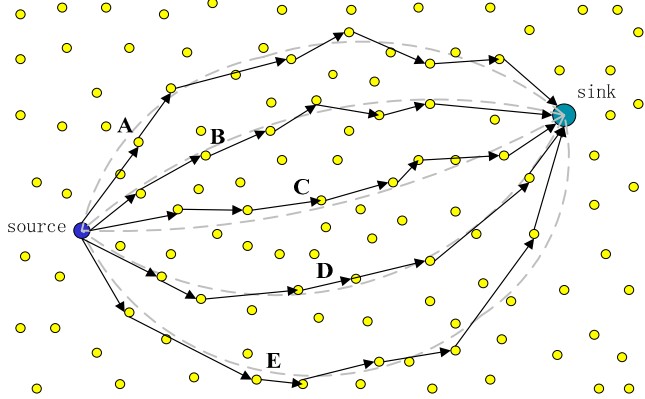

**Figure 2.** B-spline multipath schematic diagram.

### 4.1. Path Generation

Some studies have applied B-spline algorithm to wireless networks. B-spline curves can be generated by parameters of any control point ($\geq 2$), and the path length of B-spline can be calculated by function integral. It has low computational complexity and good controllability.

As shown in Figure 3, A B-spline curve can be divided into three parts: spreading, parallel, and converging. $n_s$ and $n_t$ represent the source node and sink node, respectively. Arc *sp* is controlled by points $n'_s$, $n_s$, $n_{cs}$, $n_{ct}$. Arc *pl* is controlled by points $n_s$, $n_{cs}$, $n_{ct}$, $n_t$. Arc *lt* is controlled by points $n_{cs}$, $n_{ct}$, $n_t$, $n'_t$.

$$\begin{cases} S_{sp}(t) = \frac{1}{6}n'_s(1-t)^3 + \frac{1}{6}n_s(3t^3 - 6t^2 + 4) + \frac{1}{6}n_{cs}(-3t^3 + 3t^2 + 3t + 1) + \frac{1}{6}n_{ct}t^3 \\ S_{pl}(t) = \frac{1}{6}n_s(1-t)^3 + \frac{1}{6}n_{cs}(3t^3 - 6t^2 + 4) + \frac{1}{6}n_{ct}(-3t^3 + 3t^2 + 3t + 1) + \frac{1}{6}n_t t^3 \\ S_{lt}(t) = \frac{1}{6}n_{cs}(1-t)^3 + \frac{1}{6}n_{ct}(3t^3 - 6t^2 + 4) + \frac{1}{6}n_t(-3t^3 + 3t^2 + 3t + 1) + \frac{1}{6}n'_t t^3 \end{cases} \quad (19)$$

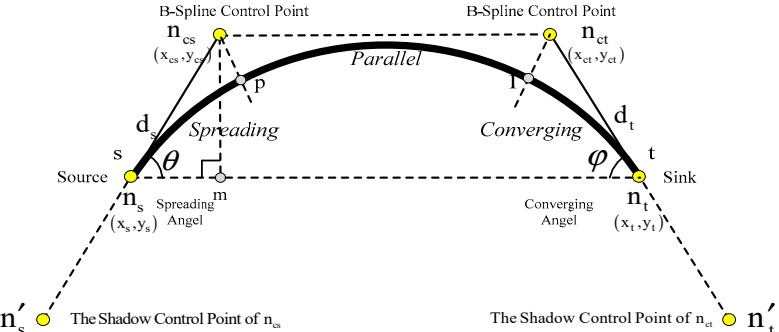

**Figure 3.** Routing based on B-spline curve.

The specific calculation is shown in the following Equation (20), which varies from 0 to 1. Note that the abscissa and ordinate coordinates in the formula are calculated separately, and the coordinate $(x_{cs}, y_{cs})$ of $n_{cs}$ is calculated by the following formula:

$$\begin{cases} x_{cs} = x_s + d_s \cdot \cos(\theta + \gamma) \\ y_{cs} = y_s + d_s \cdot \sin(\theta + \gamma) \end{cases} \tag{20}$$

where $(x_s, y_s)$ is the coordinate of node $n_s$. $\gamma = \arctan \frac{y_t - y_s}{x_t - x_s}$ is the expansion angle. $d_s$ denotes the expansion distance. It can be calculated by the following formula:

$$d_{s.} = \frac{\lambda}{\sin(\theta)} \tag{21}$$

where $n_{cs}$ and $n'_s$ are based on point $n_s$ symmetry. $n_{ct}$ and $n'_t$ are based on point $n_t$ symmetry. It can be calculated by the following formula:

$$n'_s = 2n_{cs} - n_s \tag{22}$$

In fact, only four vertices $n_s$, $n_t$, $n_{cs}$, $n_{ct}$ and this curve can be determined. The determination of $n_{cs}$ can be determined by two parameters. The first is the angle $\theta$, and the second is the length of the line from $n_{cs}$ to m. Multiple paths can be formed by setting different angles $\theta$ and the length of line segments $n_{cs}$ to $n_s$.

Assuming that $K$ path are generated, $\theta_k$ denotes the divergence angle of the $k$-th path. $\lambda_k$ denotes the maximum distance between the path $k$ and the midline. $\lambda_k$ denotes the maximum distance from the middle line of the $k$-th path. Then $\theta_k$ can be calculated by the following formula:

$$\theta_k = \Delta\theta \cdot \left| k - 1 - \frac{k-1}{2} \right| \tag{23}$$

where $\Delta\theta$ denotes the expansion angle, as shown in Figure 4. In order to distribute multipath as evenly as possible in space. The deviation angle between adjacent paths is set to the same value. In order to prevent interference between paths, the distance between adjacent paths is set to $\Delta\lambda$ as the maximum transmission distance $R_{\max}$.

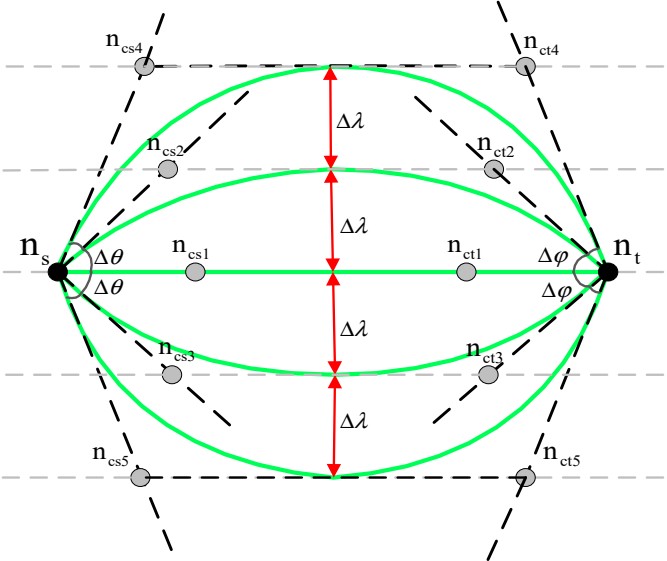

**Figure 4.** Multipath B-spline routing.

Therefore, routing can be established by three parameters, i.e., $K$, $\Delta\theta$, $\Delta\lambda \cdot \Delta\theta$ adjusts the density between paths to specify control points. By specifying the maximum expansion angle $\theta_{max}$, then $\Delta\theta$ can be calculated by the following formula:

$$\Delta\theta = \frac{2\theta_{max}}{K-1} \tag{24}$$

### 4.2. Direction Control Algorithm

The path is controlled by the expansion angle, and the expansion angle is determined by the path length. The direction control algorithm consists of two parts: calculating the length of the curve and calculating the expansion angle.

As shown in Figure 5, $n_i$ represents the current node, $n_{i+1}$ represents the next hop node. $v_i$ and $v_{i+1}$ represent virtual nodes of $n_i$ and $n_{i+1}$, respectively. $v_{i+1}$ is the ideal next hop node of $n_i$ node. The nearest node at this location is $n_{i+1}$, so $n_{i+1}$ is chosen as the next hop node. $(x_t, y_t)$ represents the coordinates of $v_i$, $\Delta L_t$ represents $v_i$ path increment. $(x_{t+\Delta t}, y_{t+\Delta t})$ denotes the next node of the spline curve. Then $\Delta L_t$ can be calculated by the following formula:

$$\Delta L_t = \sqrt{(x_{t+\Delta t} - x_t)^2 + (y_{t+\Delta t} - y_t)^2} \tag{25}$$

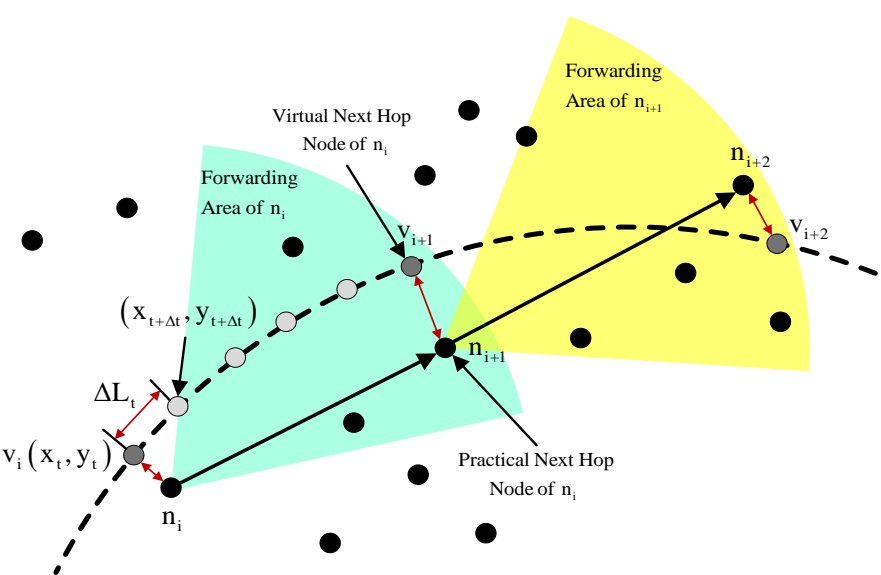

**Figure 5.** Forwarding packets to the next hop.

Cumulative, then the formula of the total length of the path can be calculated by the following formula:

$$L = \sum_{j=0}^{\frac{1}{\Delta t}} \Delta L_j \tag{26}$$

where the smaller the value of $\Delta t$, the closer the length of path calculation is to the length of ideal curve of B-spline formula. In practice, the curve length $L = GetLength(\theta, \lambda)$ is calculated by self-defining integral function.

Given $L_{max}$, $\theta_{max}$ can be calculated by Algorithm 1, and $\Delta\theta$ can be calculated by Equation (24):

---

**Algorithm 1.** Get Spreading Angle ($\tau$, $L$).

Input: $\tau$ is a small increment of $\theta$; $L$ is specified path length;

Output: $\theta$

---

1.    $\theta = 0$;
2.    $L'$=GetLength($\theta,\lambda$);
3.    while $|L' - L| > e$ do
4.      $\theta = \theta + \tau$;
5.      $L'$=GetLength($\theta,\lambda$);
6.    end while
7.    return $\theta$

---

### 4.3. Error Correction Calculation

In most cases, the next hop node is closer than the virtual node. As the node forwards, this error accumulates. Therefore, a correction factor $\rho$ is introduced to improve the actual transmission distance per hop and eliminate the error [19]. In Figure 6, $v_{i+1}$ is the ideal next hop node of $n_i$ node, $n_i$ actually chooses the next hop node as $n_{i+1}$ node. m is the center of gravity of the sector region with $v_{i+1}$ as the center and r as the radius. From a statistical point of view, the actual selection of the next hop node is close to m point. In order to calculate the position of m, we need to calculate the center of gravity of the sector area. We need to determine the size of the sector area A and the center position $v_{i+1}$. A is set here as 1/2 of the average area of the nodes. When A and $R$ are given, then r and $\alpha$ can be calculated according to the following formula:

$$\begin{cases} \cos \frac{\alpha}{2} = \frac{r}{R} \\ A \approx \frac{\alpha}{2\pi} \cdot \pi R^2 \end{cases} \tag{27}$$

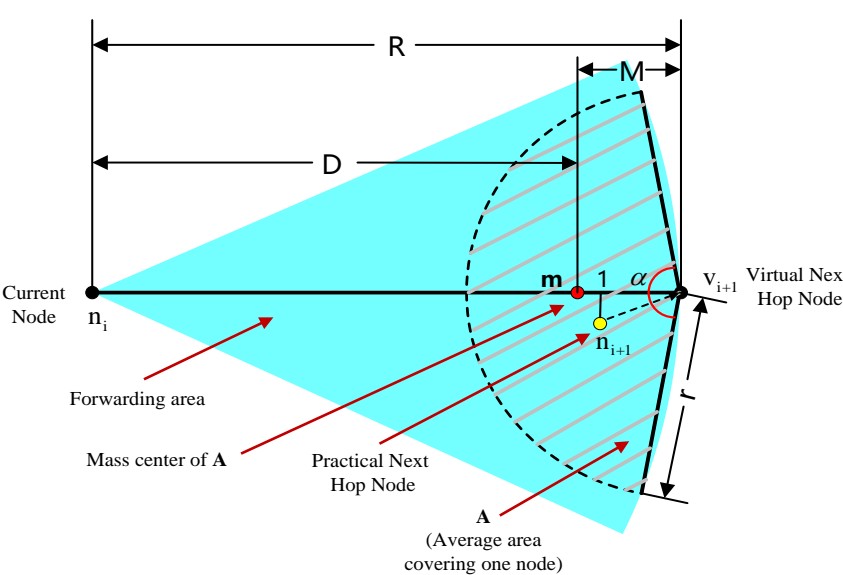

**Figure 6.** Calculating the correction factor.

If $D = R - M = R - \frac{4}{3} \cdot \frac{r}{\alpha} \cdot \sin^2 \frac{\alpha}{2}$, the formula for calculating the correction factor is as follows:

$$\rho = \frac{R}{D} = \frac{R}{R - \frac{4}{3} \cdot \frac{r}{\alpha} \cdot \sin^2 \frac{\alpha}{2}} \tag{28}$$

## 5. Implementation of Multimedia Independent Multipath Routing Algorithms

When multiple source nodes send data to the sink node at the same time, it poses a greater challenge to the construction of independent multipath routing. During the transmission process,

some source nodes may exit or join dynamically. In order to make better use of the path resources, the source nodes need to readjust the path. In this paper, the idea of software definition is applied to centralize programming and control the multi-path routing of multiple source nodes through sink nodes, so as to realize the multi-source and multi-path coordination algorithm [20,21]. There are three source nodes and one sink node in the model. Each source node sends data to sink through four paths.

Considering that sink nodes are located in the center of the sensor network and multiple source nodes send data at the same time, the following issues need to be considered.

① How do source nodes distribute their paths when the distribution of source nodes is not uniform around the sink nodes;

② How to distribute paths when the weights of source nodes are different;

How to dynamically adjust the existing multipath routing after the source node joins and exits dynamically.

### 5.1. Process Description

Message and package structure tables are shown in Table 2. Packet content includes package type, serial number, source node and target node.

**Table 2.** Algorithmic message and packet structure design.

| Package | Source | Target | Main Field | Route of Transmission |
|---|---|---|---|---|
| RREQ | Source | Sink | 1.Priority; 2.Flow; 3.Source coordinates | Shortest path |
| RACK | Sink | Source | 1. Beginning Angle; 2. Ending Angle | Shortest path |
| DATA | Source | Sink | 1. Four node coordinates that determine the path; 2. ordinal number of t; 3. subpath number; 4. data, etc. | B-spline multipath transmission |
| DATA_CANCEL | Source | Sink | Source ID | Shortest path |

### 5.1.1. Source Application Process

①After the initialization of the source node is finished, the RREQ packet is sent by setting self-interrupt trigger. In addition to the initialization phase, the network also allows nodes to send applications when it is running normally.

②All source nodes initialize RREQ packages. The fields in the packages include unit traffic, priority and coordinates of Source, which are sent by the shortest radial sink.

③After receiving RREQ packets, the sink is inserted into the source node queue from large to small according to the angle of the source relative to the sink. Set a self-interrupt because there are multiple source nodes. The interrupt time ensures that all node requests are processed only after RREQ messages have been received from all source nodes.

④The sink processes the source node queue, calculates the node weight according to a certain algorithm, assigns angles to each source node in the queue according to the weight ratio, and then sends RACK packets to all source nodes in turn.

⑤After all sources receive RACK packets, they start sending data.

### 5.1.2. Source Sends Data Flow

①Nodes begin to send data according to the starting and ending angles of applications, and the angle range applied by nodes is different in size because of different weights [22].

②The number of paths to be sent is specified according to the traffic of the node (the number of paths to be selected according to the traffic and priority of the node, such as 200,000 paths can be considered).

③According to B-spline curve algorithm, multi-radial sink sends data packets.

### 5.1.3. Source Cancels the Process

①When the source node does not want to send any more data, it sends a DATA_CANCEL message to the sink node.

②When the sink node receives it, it is removed from the source node queue. The weight and allocation angle of each source node are recalculated, and RACK packets are sent to all source nodes.

③Step 4 and step 5 of enter the source application process.

### 5.2. Algorithmic Implementation Details

#### 5.2.1. Sink End Angle Allocation Algorithms

In order to reduce the interference of wireless signals, the receiving angle range is allocated to each node according to the weight of source nodes, which makes the paths between nodes independent. First, we need to calculate the weight of each source node, the weight = traffic × priority. Then, the angle range that nodes can allocate is equal to node weight/all node weight and × 360. The formula is as follows:

$$Q_i = P_i \times F_i \tag{29}$$

$$Q_{total} = \sum_{i=1}^{n} Q_i \tag{30}$$

$$Angle_i = \frac{Q_i}{Q_{total}} \times 360 \tag{31}$$

where $P_i$ is the priority of the $i$th node, $F_i$ is the traffic of the $i$th node, $Q_i$ is the weight of the $i$th node, $Q_{total}$ is the sum of all node weights, $Angle_i$ is the angle range allocated by the $i$ th node.

#### 5.2.2. The Choice of the Order of Angle Distribution

If the distribution of source nodes is not uniform, the angles assigned may vary in size. In order to balance the load, the paths need to be evenly distributed around sink nodes, so the order of angle allocation needs to be considered. The definition of coordinate system is that the angle of the horizontal line on the right side of the node is 0, clockwise from small to large. Consider the following scenarios: one sink, three sources, namely Src1, Src2, Src3. Assume that the weights of Src1 and Src3 are 1 and that of Src2 is 2. Then there are two considerations for the allocation scheme.

①Location allocation based on the source.

The first allocation scheme is based on the location allocation of the source. The smaller the angle of Source relative to the sink, the earlier the allocation. Since the angle of Src1 is at 090, the angle range that should be allocated according to the weight calculation is as follows:

$$Angle_1 = \frac{Q_1}{Q_{total}} = 1/(1 + 2 + 1) \times 360 = 90 \tag{32}$$

With Src1 as the center, the distribution angle of Src1 is 45°–135°. The weight of Src2 is 2 and its angle range is 180. Then the angle of distribution is followed by Src1, i.e., 135°–315°. Src3 ranges from 315° to 45°, as shown in Figure 7a. Although Src1 is located in the middle of its angle range, Src2 is located on one side of its angle range, while Src3 is completely out of its angle range.

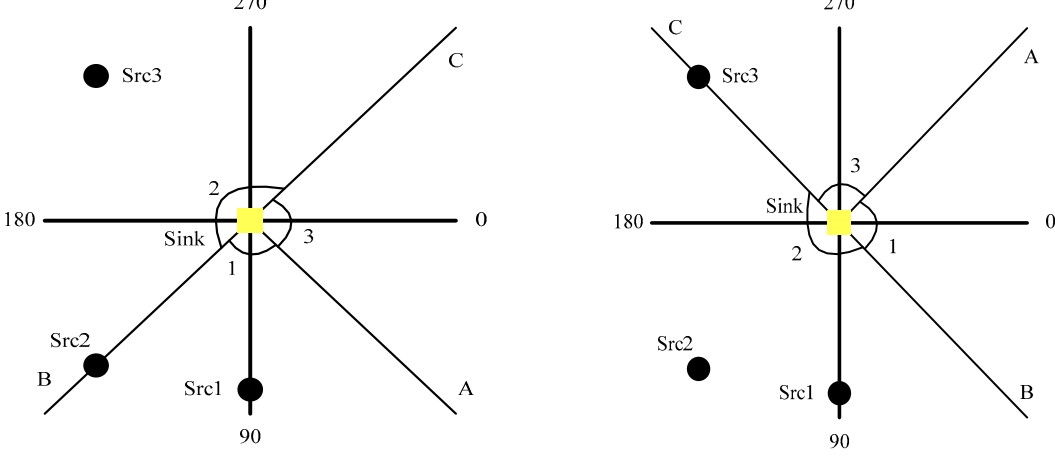

(**a**) Source location allocation graph    (**b**) Weight distribution angle

**Figure 7.** Two allocation strategies.

②Distribution begins with the node with the largest weight.

Considering that the larger the weight of the node, the larger the angle range of the allocation, in order to take care of the node with the largest weight, the node with the largest weight is assigned priority here. As shown in Figure 7b. The distribution range of Src2 is from 45° to 225° and occupies the range of Src1. The range of Src1 is from A to B, ranging from −45° to 45°.

### 5.2.3. Selection of the Sending Angle Range at the Source End

For the source, because the nodes are not located in the middle of the allocated angle range, the transmission angle range of the path should be considered. In order to keep the sending routes between source nodes from crossing, the range of the source sending angle needs to be determined according to the receiving angle allocated by the sink terminal. The formula is as follows:

$$
\begin{cases}
Angle_{Source} = Angle_{\sin k} + 180 \\
Angle_{Source\_begin} = Angle_{Source} - (Angle_{\sin k\_begin} - Angle_{\sin k}) \\
Angle_{Source\_end} = Angle_{Source} - (Angle_{\sin k\_end} - Angle_{\sin k})
\end{cases}
\tag{33}
$$

As shown in Figures 4–8, for the sink, Src1 is located at 90 degree angle of the sink, and the range of angle allocated is (−45° to 45°), while the relative angle offset is (−135° to 45°). According to Equation (33), that is, the sink is at a 270 degree angle of SRC 1. According to Equation (33), $Angle_{Source} = Angle_{Sink} + 180 = 270°$, that is, the sink is at 270 degrees of Scr1. The starting angle of Src1 is 270°−(−135°) = 405°. The end angle is 270° − (−45°) = 315°. The transmission curve is shown in Figure 8.

### 5.2.4. Determination of the Number of Paths at Source end and the Angle between Paths

If different source nodes install different camera resolutions, so the amount of data to be transmitted per unit time is different. In order to balance the paths, the size of each path packet is limited to no more than $pk_{size}$ at a time. This requires $Path_{num} = F/pk_{size}$ routes to be sent, of which $F$ represents traffic. $AngleScope_{path}$ represents the angle of interval between each path. In the edge zone, $AngleScope_{path}$ of 1/2 is set aside as the interval, and the formula for calculating the interval angle between paths is as follows:

$$
AngleScope_{path} = \left(Angle_{Source\_begin} - Angle_{souce\_end}\right)/Path_{num}
\tag{34}
$$

$$Angle_{pathi} = Angle_{Source\_begin} - i * AngleScope_{path} - 1/2 * AngleScope_{path} \tag{35}$$

For example, Src1 needs 600 KB of traffic and 200 KB of packet size in a certain period of time. Assuming that each path can fully meet the delay requirement, the required number of paths is $600k/200k = 3$. $AngleScope_{path} = (405 - 315)/3 = 30°$. According to the formula, the angles of the three paths are 390 degrees, 360 degrees, and 330 degrees, respectively.

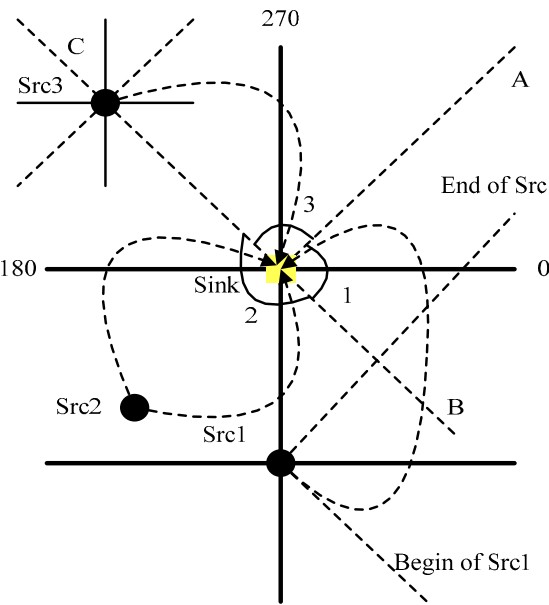

**Figure 8.** Selection of the source sending angle range.

## 6. Simulation Verification

In order to verify the proposed independent multi-path routing algorithm for networked multimedia, a simulation experiment is designed. The simulation scenario is 500 m × 900 m with 2000 nodes, and the average area of each node is 225 square meters. It is assumed that both the receiving end node and the ordinary sensor node are stationary. The sensor application module consists of a fixed bit rate data source, which generates multimedia traffic with QoS requirements. We use IEEE 802.11b DCF as the MAC layer protocol. All nodes know their geographic location by means of positioning system. All nodes know the geographic location of other nodes within their transmission radius and call these other nodes neighbors or one-hop reachable nodes.

### 6.1. Delay and Energy Model Verification

$t$ is set to 0.01 and five paths are set. Table 3 is the actual simulation results, $T_{ete}$ is the end-to-end delay, $L_e$ is the path length, $H_{ete}$ is the actual hop number, $T_{ete}/H_{ete}$ is the actual hop delay, $L_e/H_{ete}$ is the actual hop length. The expansion angle increases from 40 to 120 degrees. With the increase, $T_{ete}$ and $L_e$ increased correspondingly.

**Table 3.** Expansion angle, path length, and each distance.

| $\theta$ | $T_{ete}$ | $L_e$ | $H_{ete}$ | $T_{ete}/H_{ete}$ | $L_e/H_{ete}$ |
|---|---|---|---|---|---|
| 40 | 0.0288 | 561.23 | 11 | 0.0026 | 51.02 |
| 60 | 0.0288 | 582.84 | 11 | 0.0026 | 52.989 |
| 80 | 0.0288 | 607.94 | 11 | 0.0026 | 55.27 |
| 100 | 0.0317 | 641.33 | 12 | 0.0026 | 53.44 |
| 120 | 0.0346 | 693.69 | 13 | 0.0027 | 53.36 |
| Average | - | - | - | 0.0026 | 53.10 |

The average delay per hop is 0.0026 s, which is consistent with the theoretical calculation. It is noted that the actual distance per hop is 53 m, which is smaller than the 60 m set. Since the actual distance per hop is smaller than the corresponding virtual distance per hop. In order to verify the impact on energy consumption and time delay, the simulation is set from 25 m to 60 m, and the results are shown in Figure 9. Energy consumption and time delay show a compromise. The smaller $R_k$ is, the larger the delay and the lower the energy consumption are. Under the condition of meeting the demand of time delay, the smaller $R_k$ is, the lower energy consumption is.

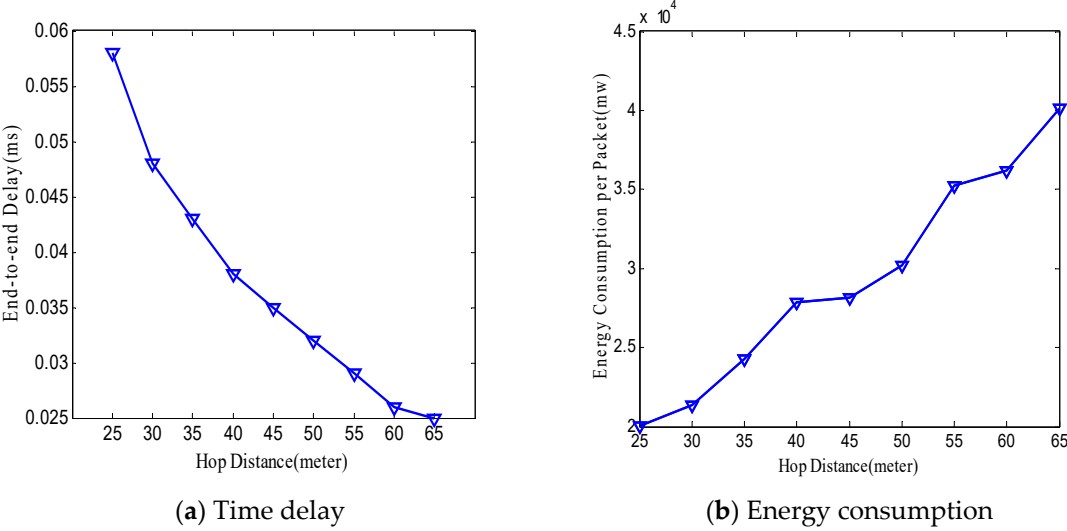

(**a**) Time delay    (**b**) Energy consumption

**Figure 9.** Delay and energy consumption.

*6.2. Performance Parameter*

Set $T_{Qos}$ to 0.03 s, and the number of paths is 5. There are three schemes: pure spline multipath, delay multipath and delay error correction multipath.

①Pure spline multipath: no delay requirement. The results are shown in Table 4. The path delay changes from 0.027 s to 0.039 s. Two of the five roads cannot meet the delay requirements. On the other hand, without energy consumption control, the maximum transmission distance is always used for each hop, and the average energy consumption is high. This scheme cannot meet the demand of time delay and has high energy consumption.

**Table 4.** Multipath performance parameters without delay requirements.

| | End-to-End Delay (s) | | | | | Energy Consumption Per Packet (mw) | | | | |
|---|---|---|---|---|---|---|---|---|---|---|
| Times (s) | Path 1 | Path 2 | Path 3 | Path 4 | Path 5 | Path 1 | Path 2 | Path 3 | Path 4 | Path 5 |
| 1 | 0.00345 | 0.0322 | 0.0284 | 0.0271 | 0.0363 | 37,200 | 26,100 | 25,400 | 28,500 | 37,200 |
| 2 | 0.00329 | 0.0278 | 0.0274 | 0.0270 | 0.0390 | 37,200 | 26,100 | 25,400 | 28,500 | 37,200 |
| 3 | 0.0356 | 0.0318 | 0.0267 | 0.0272 | 0.0387 | 37,200 | 26,100 | 25,400 | 28,500 | 37,200 |
| 4 | 0.0361 | 0.0321 | 0.0278 | 0.0271 | 0.0386 | 37,200 | 26,100 | 25,400 | 28,500 | 37,200 |
| 5 | 0.0371 | 0.0324 | 0.0279 | 0.0275 | 0.0381 | 37,200 | 26,100 | 25,400 | 28,500 | 37,200 |
| 6 | 0.0382 | 0.0327 | 0.0280 | 0.0269 | 0.0394 | 37,200 | 26,100 | 25,400 | 28,500 | 37,200 |
| 7 | 0.0346 | 0.0297 | 0.0276 | 0.0271 | 0.0388 | 37,200 | 26,100 | 25,400 | 28,500 | 37,200 |
| 8 | 0.0348 | 0.0333 | 0.0279 | 0.0271 | 0.0386 | 37,200 | 26,100 | 25,400 | 28,500 | 37,200 |
| 9 | 0.0349 | 0.0286 | 0.0265 | 0.0275 | 0.0382 | 37,200 | 26,100 | 25,400 | 28,500 | 37,200 |
| 10 | 0.0350 | 0.0327 | 0.0275 | 0.0274 | 0.0383 | 37,200 | 26,100 | 25,400 | 28,500 | 37,200 |
| 11 | 0.0351 | 0.0308 | 0.0281 | 0.0273 | 0.0378 | 37,200 | 26,100 | 25,400 | 28,500 | 37,200 |
| 12 | 0.0353 | 0.0311 | 0.0277 | 0.0272 | 0.0375 | 37,200 | 26,100 | 25,400 | 28,500 | 37,200 |
| 13 | 0.0354 | 0.0314 | 0.0274 | 0.0268 | 0.0390 | 37,200 | 26,100 | 25,400 | 28,500 | 37,200 |
| 14 | 0.0358 | 0.0305 | 0.0268 | 0.0270 | 0.0387 | 37,200 | 26,100 | 25,400 | 28,500 | 37,200 |
| 15 | 0.0379 | 0.0293 | 0.0262 | 0.0271 | 0.0392 | 37,200 | 26,100 | 25,400 | 28,500 | 37,200 |
| 16 | 0.0363 | 0.0301 | 0.0274 | 0.0263 | 0.0396 | 37,200 | 26,100 | 25,400 | 28,500 | 37,200 |
| 17 | 0.0358 | 0.0323 | 0.0283 | 0.0268 | 0.0378 | 37,200 | 26,100 | 25,400 | 28,500 | 37,200 |
| 18 | 0.0349 | 0.0294 | 0.0279 | 0.0271 | 0.0375 | 37,200 | 26,100 | 25,400 | 28,500 | 37,200 |
| 19 | 0.0352 | 0.0303 | 0.0273 | 0.0269 | 0.0372 | 37,200 | 26,100 | 25,400 | 28,500 | 37,200 |
| 20 | 0.0354 | 0.0311 | 0.0282 | 0.0271 | 0.0374 | 37,200 | 26,100 | 25,400 | 28,500 | 37,200 |

②Delay multipath: End-to-end delay is generally within the delay requirement, and energy consumption is also smaller than the previous scheme, as shown in Table 5.

**Table 5.** Performance parameters of the delay multipath.

| | End-to-End Delay (s) | | | | | Energy Consumption per Packet (mw) | | | | |
|---|---|---|---|---|---|---|---|---|---|---|
| Time (s) | Path 1 | Path 2 | Path 3 | Path 4 | Path 5 | Path 1 | Path 2 | Path 3 | Path 4 | Path 5 |
| 1 | 0.0345 | 0.0323 | 0.0316 | 0.0321 | 0.0357 | 32,800 | 22,400 | 18,600 | 22,200 | 31,500 |
| 2 | 0.0346 | 0.0318 | 0.0324 | 0.0320 | 0.0350 | 32,800 | 22,400 | 18,600 | 22,200 | 31,500 |
| 3 | 0.0356 | 0.0305 | 0.0308 | 0.0302 | 0.0361 | 32,800 | 22,400 | 18,600 | 22,200 | 31,500 |
| 4 | 0.0346 | 0.0325 | 0.0328 | 0.0316 | 0.0345 | 32,800 | 22,400 | 18,600 | 22,200 | 31,500 |
| 5 | 0.0351 | 0.0329 | 0.0314 | 0.0305 | 0.0338 | 32,800 | 22,400 | 18,600 | 22,200 | 31,500 |
| 6 | 0.0342 | 0.0303 | 0.0310 | 0.0316 | 0.0362 | 32,800 | 22,400 | 18,600 | 22,200 | 31,500 |
| 7 | 0.0346 | 0.0321 | 0.0306 | 0.0317 | 0.0347 | 32,800 | 22,400 | 18,600 | 22,200 | 31,500 |
| 8 | 0.0348 | 0.0320 | 0.0319 | 0.0304 | 0.0352 | 32,800 | 22,400 | 18,600 | 22,200 | 31,500 |
| 9 | 0.0343 | 0.0305 | 0.0305 | 0.0324 | 0.0348 | 32,800 | 22,400 | 18,600 | 22,200 | 31,500 |
| 10 | 0.0355 | 0.0317 | 0.0331 | 0.0328 | 0.0353 | 32,800 | 22,400 | 18,600 | 22,200 | 31,500 |
| 11 | 0.0356 | 0.0307 | 0.0306 | 0.0313 | 0.0344 | 32,800 | 22,400 | 18,600 | 22,200 | 31,500 |
| 12 | 0.0346 | 0.0321 | 0.0311 | 0.0302 | 0.0357 | 32,800 | 22,400 | 18,600 | 22,200 | 31,500 |
| 13 | 0.0352 | 0.0307 | 0.0307 | 0.0327 | 0.0338 | 32,800 | 22,400 | 18,600 | 22,200 | 31,500 |
| 14 | 0.0351 | 0.0311 | 0.0334 | 0.0320 | 0.0346 | 32,800 | 22,400 | 18,600 | 22,200 | 31,500 |
| 15 | 0.0349 | 0.0313 | 0.0315 | 0.0311 | 0.0353 | 32,800 | 22,400 | 18,600 | 22,200 | 31,500 |
| 16 | 0.0354 | 0.0326 | 0.0316 | 0.0327 | 0.0336 | 32,800 | 22,400 | 18,600 | 22,200 | 31,500 |
| 17 | 0.0351 | 0.0308 | 0.0318 | 0.0331 | 0.0354 | 32,800 | 22,400 | 18,600 | 22,200 | 31,500 |
| 18 | 0.0346 | 0.0324 | 0.0306 | 0.0306 | 0.0365 | 32,800 | 22,400 | 18,600 | 22,200 | 31,500 |
| 19 | 0.0350 | 0.0331 | 0.0323 | 0.0309 | 0.0339 | 32,800 | 22,400 | 18,600 | 22,200 | 31,500 |
| 20 | 0.0347 | 0.0314 | 0.0312 | 0.0305 | 0.0354 | 32,800 | 22,400 | 18,600 | 22,200 | 31,500 |

③Delay error correction multipath: When the transmission distance of each path is specified, the value of A is half of the average area of each node, that is 112.5 square meters.

It is observed that the transmission distance is inversely proportional to the transmission distance, and the actual transmission distance per hop is corrected. Higher accuracy is obtained. End-to-end delay is guaranteed and more energy is saved. As shown in Table 6.

**Table 6.** Performance parameters without delay requirements.

| Time (s) | End-to-End Delay (s) | | | | | Energy Consumption per Packet (mw) | | | | |
|---|---|---|---|---|---|---|---|---|---|---|
| | Path 1 | Path 2 | Path 3 | Path 4 | Path 5 | Path 1 | Path 2 | Path 3 | Path 4 | Path 5 |
| 1 | 0.0287 | 0.0282 | 0.0294 | 0.0292 | 0.0301 | 28,200 | 25,500 | 22,800 | 23,700 | 27,800 |
| 2 | 0.0289 | 0.0283 | 0.0278 | 0.0280 | 0.0290 | 28,200 | 25,500 | 22,800 | 23,700 | 27,800 |
| 3 | 0.0281 | 0.0302 | 0.0284 | 0.0322 | 0.0283 | 28,200 | 25,500 | 22,800 | 23,700 | 27,800 |
| 4 | 0.0290 | 0.006 | 0.0312 | 0.0321 | 0.0276 | 28,200 | 25,500 | 22,800 | 23,700 | 27,800 |
| 5 | 0.0288 | 0.0284 | 0.0288 | 0.0309 | 0.0289 | 28,200 | 25,500 | 22,800 | 23,700 | 27,800 |
| 6 | 0.0302 | 0.0297 | 0.0320 | 0.0288 | 0.0293 | 28,200 | 25,500 | 22,800 | 23,700 | 27,800 |
| 7 | 0.0306 | 0.0293 | 0.0312 | 0.0273 | 0.0288 | 28,200 | 25,500 | 22,800 | 23,700 | 27,800 |
| 8 | 0.0281 | 0.0286 | 0.0287 | 0.0291 | 0.0386 | 28,200 | 25,500 | 22,800 | 23,700 | 27,800 |
| 9 | 0.0287 | 0.0282 | 0.0301 | 0.0287 | 0.0292 | 28,200 | 25,500 | 22,800 | 23,700 | 27,800 |
| 10 | 0.0291 | 0.0307 | 0.0298 | 0.0284 | 0.0303 | 28,200 | 25,500 | 22,800 | 23,700 | 27,800 |
| 11 | 0.0296 | 0.0308 | 0.0288 | 0.0278 | 0.0288 | 28,200 | 25,500 | 22,800 | 23,700 | 27,800 |
| 12 | 0.0284 | 0.0301 | 0.0307 | 0.0282 | 0.0300 | 28,200 | 25,500 | 22,800 | 23,700 | 27,800 |
| 13 | 0.0289 | 0.0284 | 0.0324 | 0.0298 | 0.0312 | 28,200 | 25,500 | 22,800 | 23,700 | 27,800 |
| 14 | 0.0288 | 0.0295 | 0.0308 | 0.0293 | 0.0302 | 28,200 | 25,500 | 22,800 | 23,700 | 27,800 |
| 15 | 0.0308 | 0.0298 | 0.0298 | 0.0289 | 0.0282 | 28,200 | 25,500 | 22,800 | 23,700 | 27,800 |
| 16 | 0.0303 | 0.0271 | 0.0284 | 0.0293 | 0.0293 | 28,200 | 25,500 | 22,800 | 23,700 | 27,800 |
| 17 | 0.0298 | 0.0303 | 0.0303 | 0.0287 | 0.0279 | 28,200 | 25,500 | 22,800 | 23,700 | 27,800 |
| 18 | 0.0302 | 0.0284 | 0.0289 | 0.0314 | 0.0285 | 28,200 | 25,500 | 22,800 | 23,700 | 27,800 |
| 19 | 0.0282 | 0.0302 | 0.0288 | 0.0319 | 0.0302 | 28,200 | 25,500 | 22,800 | 23,700 | 27,800 |
| 20 | 0.0284 | 0.0291 | 0.0295 | 0.0289 | 0.0311 | 28,200 | 25,500 | 22,800 | 23,700 | 27,800 |

*6.3. Performance Parameter*

Most of the existing multimedia independent multipath routing algorithms are based on the maximum transmission distance per hop, without considering the balance of delay and energy consumption. Therefore, it is necessary to study new delay and energy balance routing algorithms. In this paper, a new algorithm is proposed. When multiple source nodes send data to sink nodes through multiple paths at the same time, in order to avoid interference caused by path crossover, Multiple source nodes coordinate transmission paths to avoid interference. We compare the algorithm with RTGOR algorithm [11] and TIGMR algorithm [12], and compare the following parameters:

①Average end-to-end delay, including various delays, such as queuing, retransmitting and transmission delays.

②Energy consumption E (mw) per round: Total energy consumption of 60 packets in all routes:

$$E = \sum_{k=1}^{K} \sum_{i=1}^{P_k} (D_{n_{i-1}n_i})^2 \approx \sum_{k=1}^{K} \sum_{i=1}^{P_k} P_k{}^2 \tag{36}$$

A large number of simulations have been done to change $T_{Qos}$ from 0.02 s to 0.04 s. Figure 10a shows the time-delay comparison. Figure 10b shows the energy consumption comparison of several routing algorithms. Since the time-delay is not considered in RTGOR algorithm and TIGMR algorithm, the energy consumption of the proposed algorithm is higher. The energy consumption of the proposed algorithm is better than that of the algorithm in RTGOR algorithm and TIGMR algorithm, and its energy consumption increases with the increase of $T_{Qos}$. It is necessary to set $T_{Qos}$ according to the actual situation in order to give full play to the advantages of the algorithm. When the number of paths increases from three to seven, because of the interference between paths and various network attacks, the energy consumption and delay of the three algorithms increase, because the path of the TIGMR algorithm is denser than other algorithms, so the delay is lower. The best performance of this algorithm is to satisfy the requirement of QoS delay. The interference between the path and various network attacks is avoided to the greatest extent, and the energy consumption is the lowest.

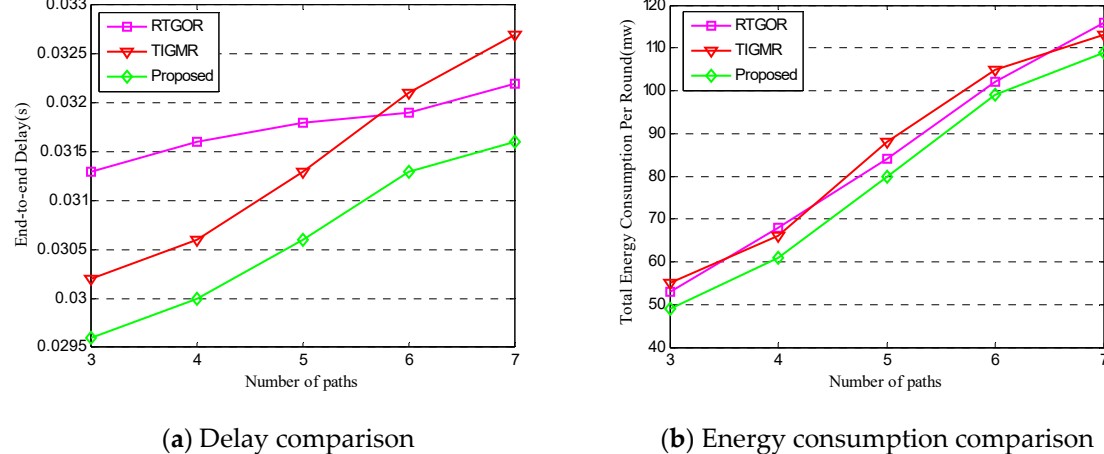

(**a**) Delay comparison　　　　　　　　　　　　　　(**b**) Energy consumption comparison

**Figure 10.** Time delay and energy consumption comparison of several methods.

## 7. Concluding Remarks

IoT is an extension of the Internet, which extends the scope of interconnection to anything and everything. The new generation of information and network technology with IoT and software defined networks as the core is the hotspot of current computer network research.

This paper studies the multi-media independent multi-path routing algorithm. Using the idea of software definition, a multi-source multi-path routing algorithm is proposed. Through sink node centralized programming control source node routing, one can dynamically coordinate the multipath routing of each source node according to the change of the source node. The model is deeply analyzed, and the anti-attack technology based on the hidden communication model of the IoT is emphatically studied. In the node hidden communication model, the anonymity of forwarding network, sender and receiver determines the anonymity of the model. In the aspect of multimedia independent multipath routing, most of the existing multimedia independent multipath routing algorithms are based on the maximum transmission distance per hop, without considering the balance of the delay and energy consumption.

There have been many studies on multipath routing in wireless sensor networks. In the research of multipath routing, the mobility of nodes, cross-layer design, duty cycle of nodes, and data fusion need to be further studied. In addition, the energy consumption of network nodes will be further balanced in the future, ensuring the effectiveness and security of routing nodes.

**Author Contributions:** Data curation, C.W. and J.Y.; funding acquisition, C.W.; investigation: C.W.; methodology: C.W. and J.Y.; software: C.W. and J.Y.; visualization: C.W.

**Funding:** This research was funded by the Henan Science and Technology Department project (no. 182102311126), basic research project of Henan Education Department (no. 16A520106).

**Conflicts of Interest:** The authors declare no conflict of interest.

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
