# Peer review of "Multimedia Independent Multipath Routing Algorithms for Internet of Things Based on a Node Hidden Communication Model"

_futureinternet, doi:10.3390/fi11110240_

Round 1
Reviewer 1 Report
I would suggest minor revision on the basis of the following points:
On line 39 you said "researchers have adopted..." I think you should state that which researcher and in which paper they have adopted this method. It will give the proper credit to author and provide reference for readers. On line number 238 you made a spelling mistake; "n this situation" it must be "In this situation.
Other than that overall paper is good, you identified the problem and provide the solution. You proposed the new algorithm with enough mathematical support and experimental result. You have shown different graphs and tables which represents your findings and results. You have also provide the comparison of your algorithm with previously proposed algorithms and showing that your's algorithm outperform the others both in case of delay and energy consumption.
Author Response
Comment:
I would suggest minor revision on the basis of the following points:
On line 39 you said "researchers have adopted..." I think you should state that which researcher and in which paper they have adopted this method. It will give the proper credit to author and provide reference for readers. On line number 238 you made a spelling mistake; "n this situation" it must be "In this situation.
Other than that overall paper is good, you identified the problem and provide the solution. You proposed the new algorithm with enough mathematical support and experimental result. You have shown different graphs and tables which represents your findings and results. You have also provide the comparison of your algorithm with previously proposed algorithms and showing that your's algorithm outperform the others both in case of delay and energy consumption.
Response: Thanks for your comments.
Correction has been made according to your review comments.
Reviewer 2 Report
The paper proposes a multi-source multi-path routing algorithm targeted to IoT networks.
In general, the paper is well-organized and there is a wide formalization of system model and proposed approach.
Some concerns are the followings:
It is not clear why the target of the paper seems to be IoT (from the title and the abstract) and, then, the authors mainly refer to WMSN. Sections 4 and 5 must be supported by sequence diagrams or something similar, in order to clearly understand the data flow. Table 2 is difficult to read. Which tool has been used for simulation? Please provide more details in this point. Which is the metric for power consumption? Please specify the unit of measure. Conclusion must be rewritten in a better form, since such a section is difficult to read and also presentes some incorrect sentences.
Author Response
Comment:
The paper proposes a multi-source multi-path routing algorithm targeted to IoT networks.
In general, the paper is well-organized and there is a wide formalization of system model and proposed approach.
Some concerns are the followings:
It is not clear why the target of the paper seems to be IoT (from the title and the abstract) and, then, the authors mainly refer to WMSN. Sections 4 and 5 must be supported by sequence diagrams or something similar, in order to clearly understand the data flow. Table 2 is difficult to read. Which tool has been used for simulation? Please provide more details in this point. Which is the metric for power consumption? Please specify the unit of measure. Conclusion must be rewritten in a better form, since such a section is difficult to read and also presentes some incorrect sentences.
Response: Thanks for your comments.
Wireless multimedia sensor network (WMSN) belongs to the Multimedia data perception layer of the Internet of things, which has been introduced in the introduction. And there are specific steps in sections 4 and 5; Table 2 shows the algorithm message and package structure design. The form of the conclusion has been adjusted according to your suggestion
Reviewer 3 Report
Comment on internet of thing paper
This papers lacks scientific and engineering basis/artifacts especially for practical Wireless multimedia sensor network (WMSNs)
There is no novelty or contribution in the study
I do not understand the abstract
“On the premise of satisfying the application delay, a multi-source 13 multi-path routing algorithm is proposed by using the idea of software definition and fitting 14 multiple curves to form independent multi-path routing”
How?
“Through Sink node centralized 15 programming control source node routing, according to the priority of the source node, the 16 dynamic angle of the source node can be allocated, which effectively reduces the energy 17 consumption of the network”
How? It is not engineering (or scientific)
Also how does privacy protection solve the problem of delay and energy consumption?
the application of multipath routing is not new, even the traditional networks- WAN, WSN employs that
3.most of the equations are not related to the problem the author is trying to solve and there is no sequential flow of the article
Like in
3.3 transmission model, I do not understand the purpose of the model (its usefulness??),
3.2, energy will change with distance, it is not new, what is the model trying to tell us?
What is the purpose of node hidden communication model? How is encryption related to hidden mode? What kind encryption transformation is equations 14 and 15?
3.1 delay model is not new, see computer network top down approach by Kurose
section
4. What defines the path? how do you determine the length? How do you generate multipath routing in real life node? And why the trouble?
5. There are some assumptions or technicalities relating to network structure layers of any communication network that are missing
You cannot route if you do not belong to the network per region, and therefore, you cannot route without knowing the node and network information such node as node id/address, routing table information, time-to-live. The approach discussed here is not broadcasting. All these details/parameters are not shown or related to the geometric derivation of B-spline algorithm-(it is also available in text, not new).
Is node deployed in deterministic or stochastic manner, because they will have different behavior to routing because of node location per area of deployment? Just like the figures 2 and 4 appear to have different node placement/arrangement
6. I don’t understand table 2. It is wired and messy and I don’t understand it purpose
7. Some of the theories discussed here will not be applicable to real life sensor network deployment and routing
8. Section 6
IEEE 466 802.11b DCF as the MAC layer protocol is infrastructure network protocol meaning, you need IP address and it’s most of the time used in where there is not energy constraint system unlike non-infrastructure network-e.g wireless sensor network (like ZigBee, LoRawan etc) and I’m not sure how you are implement the so called proposed method on top of IEEE 466 802.11b, and where? How?
Author Response
Reviewer 3
Comment:
This papers lacks scientific and engineering basis/artifacts especially for practical Wireless multimedia sensor network (WMSNs)
There is no novelty or contribution in the study
Response: Thanks for your comments.
The main contributions of this paper are summarized as follows.
1)On the premise of satisfying the application delay, using the idea of software definition, the independent multi-path routing is formed by fitting multiple curves. A multi-source multi-path routing algorithm is proposed, which takes into account the balance of delay and energy consumption.
2)Aiming at the problem of privacy disclosure of communication relationship, a node hidden communication model is designed for the IoT. In this covert communication system, the attack detection and prevention technology is studied, which effectively improves the anti-attack ability.
The simulation experiment also proves that the proposed algorithm can avoid the interference between paths and all kinds of network attacks to the maximum extent, and the energy consumption is relatively low.
I do not understand the abstract
“On the premise of satisfying the application delay, a multi-source 13 multi-path routing algorithm is proposed by using the idea of software definition and fitting 14 multiple curves to form independent multi-path routing”
How?
Response: Thanks for your comments.
We have reorganized the abstract, reorganized the proposed method and enhanced readability.
“Through Sink node centralized 15 programming control source node routing, according to the priority of the source node, the 16 dynamic angle of the source node can be allocated, which effectively reduces the energy 17 consumption of the network”
How? It is not engineering (or scientific)
Response: Thanks for your comments.
There are specific implementation steps in Section 5.2” Algorithmic Implementation Details” at the beginning of line 401, which explains the process in detail.
Also how does privacy protection solve the problem of delay and energy consumption?
Response: Thanks for your comments.
The delay model and energy consumption model in this paper are used to solve the problems of delay and energy consumption. The hidden communication model of nodes helps to improve the level of privacy protection of the Internet of things, and effectively improves the ability of nodes to resist attacks in the Internet of things.
the application of multipath routing is not new, even the traditional networks- WAN, WSN employs that
Response: Thanks for your comments.
The innovation of this article has been summarized in the introduction:
The main contributions of this paper are summarized as follows.
1)On the premise of satisfying the application delay, using the idea of software definition, the independent multi-path routing is formed by fitting multiple curves. A multi-source multi-path routing algorithm is proposed, which takes into account the balance of delay and energy consumption.
Aiming at the problem of privacy disclosure of communication relationship, a node hidden communication model is designed for the IoT. In this covert communication system, the attack detection and prevention technology is studied, which effectively improves the anti-attack ability.
3.most of the equations are not related to the problem the author is trying to solve and there is no sequential flow of the article
Response: Thanks for your comments.
We disagree with this comment. All the formulas are useful. These formulas are built models and used to build simulation environment. How can we say they are useless.
Like in 3.3 transmission model, I do not understand the purpose of the model (its usefulness??),
3.2, energy will change with distance, it is not new, what is the model trying to tell us?
What is the purpose of node hidden communication model? How is encryption related to hidden mode? What kind encryption transformation is equations 14 and 15?
Response: Thanks for your comments.
These models support our proposed algorithm, which is certainly not useless. For example, the energy consumption model is to reduce energy consumption. The same formula (14) and formula (15) are not useless, and their functions are described in lines 194-204.
3.1 delay model is not new, see computer network top down approach by Kurose
Response: Thanks for your comments.
The method in this paper synthesizes several models instead of considering the delay problem alone.
section 4 What defines the path? how do you determine the length? How do you generate multipath routing in real life node? And why the trouble?
Response: Thanks for your comments.
These problems have been illustrated in Fig. 3 and Fig. 4, and also explained in Section 4.1, as explained in line 303: in order to distribute multipaths as evenly as possible in space, the deviation angle between adjacent paths is set to the same value.
5.There are some assumptions or technicalities relating to network structure layers of any communication network that are missing
You cannot route if you do not belong to the network per region, and therefore, you cannot route without knowing the node and network information such node as node id/address, routing table information, time-to-live. The approach discussed here is not broadcasting. All these details/parameters are not shown or related to the geometric derivation of B-spline algorithm-(it is also available in text, not new).
Is node deployed in deterministic or stochastic manner, because they will have different behavior to routing because of node location per area of deployment? Just like the figures 2 and 4 appear to have different node placement/arrangement
Response: Thanks for your comments.
For specific technical details, Chapter 5 of this paper specifically lists the detailed implementation steps, and explains each step, such as: 5.2.2. The Choice of the Order of Angle Distribution; 5.2.3. Selection of Sending Angle Range at Source End;
6. I don’t understand table 2. It is wired and messy and I don’t understand it purpose.
Response: Thanks for your comments.
We have rearranged Table 2.
7.Some of the theories discussed here will not be applicable to real life sensor network deployment and routing.
Response: Thanks for your comments.
At present, the algorithm proposed in this paper has been implemented in the simulation phase, and has achieved good results. The next step is to be applied in practice.
8.Section 6
IEEE 466 802.11b DCF as the MAC layer protocol is infrastructure network protocol meaning, you need IP address and it’s most of the time used in where there is not energy constraint system unlike non-infrastructure network-e.g wireless sensor network (like ZigBee, LoRawan etc) and I’m not sure how you are implement the so called proposed method on top of IEEE 466 802.11b, and where? How?
Response: Thanks for your comments.
At present, the algorithm proposed in this paper has been implemented in the simulation phase, and has achieved good results. The next step is to be applied in practice.
Round 2
Reviewer 2 Report
The Reviewer has no further comment.